# Analysis of High Performance Concrete Mixed with Nano-Silica in Front of Sulfate Attack

**DOI:** 10.3390/ma15217614

**Published:** 2022-10-29

**Authors:** Lianfei Nie, Xiangdong Li, Jing Li, Baolong Zhu, Qi Lin

**Affiliations:** 1China Aerodynamics Research and Development Center, Mianyang 621000, China; 2School of Civil Engineering and Architecture, Southwest University of Science and Technology, Mianyang 621010, China

**Keywords:** nano-silica, high performance concrete, sulfate attack, microstructure, compressive strength, splitting strength

## Abstract

Nano-silica (NS) is an effective material to improve the strength and durability of high-performance concrete (HPC), but little information is available regarding its role in HPC response to long-term sulfate attack. In this study, six different dosages of NS (0%, 1%, 2%, 3%, 4%, and 5%) as cement partial replacement were mixed into HPC and the casted specimens were soaked in sulfate solution for different periods (0, 100, 200, and 300 days). The mass change, dynamic elastic modulus, compressive and splitting strength, microstructure morphology, and porosity characteristics of HPC specimens were measured by mass tests, mechanical properties tests, scanning electron microscopy (SEM), and nuclear magnetic resonance (NMR) tests. The results showed that the incorporation of NS decreased the mass loss, elevated the compressive and splitting strength, and reduced the porosity formation of HPC in front of sulfate attack. The percentage of 1% NS was among the most effective dosages as, after soaking for 300 days, it decreased the mass loss by 13.5%, elevated the elastic modulus as well as compressive and splitting strength by 50.4%, 31.7%, and 69.8% in comparison of unmodified HPC, respectively. The sulfate attack resistance was delayed in a higher (2–5%) mixed dosage, mainly due to the agglomeration of nano particles, especially after long-term reactions. This study can provide experimental references regarding the performance of HPC mixed with NS in front of sulfate attack.

## 1. Introduction

High performance concrete (HPC) is a new type of concrete with the advantages of high durability, high strength, and high working performance, which is widely used in highway tunnels, underground passages, and coastal constructions [1,2,3]. In such a harsh environment, the HPC structures are continuously exposed to sulfate attack, which is well known to damage the physical and mechanical properties of concrete [4,5,6,7,8]. By penetrating to the porosities, sulfate ions can react with the hydration of cement such as calcium hydroxide, calcium silicate hydrates and calcium aluminate hydrates, then crystallized to form gypsum, ettringite and thaumasite [5,9,10]. The consumption of cement products and the formation of expansive crystallization led to the degradation and cracking of concrete, finally reduce its strength and durability [5,11,12,13]. According to ASTM recommendations [14,15,16], the sulfate resistance of Portland cements is determined by the expansion at 28 days, or at 14 days-age for Portland sulfate resistant cements, as the sulfate erosion resistance of concrete is predominately based on the chemical content and reactions of tricalcium aluminate (C_3_A) [17,18]. However, in field engineering applications, HPC constructions will continuously face exposure to a circumstance in front of sulfate attack [8,19]. Therefore, to maintain the strength and durability of HPC structures, it is necessary to thoroughly understand the physical and mechanical performance of HPC, especially under long-term influence.

The incorporation of nanomaterials has been proven to improve the physical, chemical, and mechanical properties of HPC [1,20,21,22,23,24]. Due to its small particle size, high activity, and large specific surface area, the fine nano particles can provide nucleation sites for early hydration with calcium hydroxide (CH) and secondary calcium silicate hydrate (CSH) gel of concrete, and these nano particles can also migrate into the interfacial transition zone, where is the weakest link in concrete [22,25,26]. Therefore, the pozzolanic activity and filling effect of nanomaterials will significantly enhance the microstructure and improve the mechanical specifications and erosion resistance of concrete [3,25,26,27,28]. On the other hand, the surface electrical charge of nanoparticles can regulate C_3_A hydration. Wang et al. [29] and Solanki et al. [30] revealed that the negatively charged nano-silica particles can be attracted to the positive-charged aluminate phase, then block the active dissolution front of C_3_A and retard the hydration. Thus, the nano-silica particles potentially hindered the sulfate attack and enhanced the durability of HPC. For instance, Hou et al. investigated the effect of Nano-silica on the performance of C_3_A gypsum systems under sulfate attack and observed that NS addition effectively enhanced the sulfate attack resistivity of C_3_A gypsum system. The filling effect of nano particles and the consequent reduction of morphology induced expansion were attributed to the enhancement of sulfate attack resistivity [31]. Swaidani et al. observed that the crystals of ettringite were much smaller in concrete with 3% nano volcanic scoria than in plain concrete after exposure to 5% Na_2_SO_4_ for 52 weeks [32]. Guo et al. studied the effect of nano-silica on the sulfate attack resistance of the C_3_S hydration system, and found that NS prevented the expansive transformation of calcium hydroxide into gypsum, then retarded the sulfate attack process, and the 3% NS was the most effective [28].

Previous studies indicated that the different types of nanoparticles and mix dosage influence the performance of concrete. For instance, Mokhtar et al. reported that the incorporation of nano-clay, nano-silica, and hybrid particles can elevate the compressive, splitting tensile, and flexural strength of HPC. In particular, the mixes with a cement content of 350 kg/m^3^, 6% NC, and 3% NS addition demonstrated the highest enhancement after curing for 90 days [33]. Jeon I.K. et al. reported that the incorporation of 1% and 2% nano-silica in underwater concrete improved its compressive strength and elevated the resistance to carbonation and chloride penetration after HCl erosion by 90 days [25]. Norhasri et al. reported that the nano-clay refined the micro-surface of UHPC, and enhanced its compressive strength, this enhancement was the greatest at the mixed dosage of 1%. Moreover, the micro-surface of high content of NC (3–9%) was rougher than that in 1%, and delayed the compressive strength, with the effect augmented over time [3]. Generally, such a high content of nanomaterial-induced reductions are attributed to the agglomeration of nanoparticles as, due to the high inter-particle van der Waal’s forces, the relatively high content of nanoparticles were prone to agglomerate [22,26,34]. Hou et al. observed more un-hydrated cement particles in mortar containing 5% colloidal NS than that of 2.25% [31]. Therefore, it is mandatory to intensively study the added dosage of nanomaterials for field engineering applications.

To further investigate the role of nanomaterials in the HPC response to long-term sulfate attack, in this study, six different concentrations of nano-silica (NS) (0%, 1%, 2%, 3%, 4%, and 5% wt replace of cement) were added into HPC to form cubic and cylindrical specimens. They were then soaked in sulfate solution (5% NaSO_4_) for a pre-determined period. The mass change, dynamic elastic modulus, compressive and splitting strength, microstructure morphology, and porosity characteristics of HPC specimens were measured by mass tests, mechanical properties tests, scanning electron microscopy (SEM), and nuclear magnetic resonance (NMR) tests. The primary target of this study was to investigate the effect of NS on the performance of HPC in front of long-term sulfate attack, as well as to probe the optimal content of NS applied in HPC. This study will provide experimental references to the filed application of HPC in front of sulfate attack.

## 2. Materials and Methods

### 2.1. Materials and Mix Proportions

In this study, the compositions of HPC were as follows: ordinary Portland cement (early strength type) of Grade 42.5R (Four Star Cement Co., Ltd., Mianyang, China) was used as the cement, the chemical composition of cement, fly ash, and silica fume measured by XRF, and the potential mineral composition calculated according to the Bouge’s formulas are exhibited in Table 1. Nano-silica (NS) with particle size of 20 nm (Shanghai Maikun Chemical Co., Ltd., Shanghai, China) was used, and its properties are exhibited in Table 2. Quartz sand (16–70 mesh) was used as the fine aggregate, and running water was used for the mixing and curing of HPC. Hence, 5 mix proportions (1%, 2%, 3%, 4%, 5%) of NS replacing the cement were doped internally in HPC, and the specific parameters are shown in Table 3.

### 2.2. Specimen Preparation

To avoid the formation of nanomaterial agglomeration in the cement matrix, the NS particles were firstly mixed with quartz sand and stirred at medium speed in a mixer for 120 min. Then, the cement was added to the mix. After full mixing, the water and super-plasticizer were added into the mix and stirred at a low speed.

The mixture was poured into a cylindrical mold with a size of φ50 mm × 50 mm for NMR tests, and a cubic mold with a size of 100 mm × 100 mm × 100 mm for mechanical properties measurements. After natural curing for 24 h, the specimens were removed from the molds and cured at a standard curing room (20 ± 2 °C and >95% relative humidity) for 28 days. In each mix dosage (0%, 1%, 2%, 3%, 4%, 5%), there were 3 cube specimens and 3 cylinder specimens prepared for different soaking periods (0, 100, 200, and 300 days), i.e., the measurements were performed in triplicate at each dosage and time point, and 144 specimens were cast in total, as shown in Figure 1.

### 2.3. Sulfate Attack Experiment

The HPC specimens were transferred into containers for the sulfate erosion experiment. Sodium sulfate solution with a concentration of 5% NaSO_4_ by mass was chosen as the erosion solution, according to ASTM recommendations [14,15]. The soaking periods were set as 0, 100, 200, and 300 days, respectively.

### 2.4. Measurements

#### 2.4.1. Mass Test

The initial mass of every specimen after curing was weighed as *m*_0_. After sulfate erosion of predetermined period (0, 100, 200, 300 days), specimens were removed, with their surfaces wiped dried, then dried in an oven at 105 °C for 24 h, and the mass was weighed as *m_t_*. The ME10 analytical balance (Mettler Toledo, Shanghai, China) with accuracy of 0.01 was utilized, and the mass change rate (*W*) of specimen was calculated as follows,
(1)$$W=\frac{m_{t}-m_{0}}{m_{0}}\times 100\%$$

#### 2.4.2. Elastic Modulus Test

The elastic modulus of the cube specimens was tested using a HS-YS301T acoustic tester (Hongshen, Hunan, China). In the test, the propagation velocity of ultrasonic wave in the tested material is calculated by the wave amplitude, initial arrival time, and the known distance of material, so as to obtain the dynamic elastic modulus. Using density and measured p-wave and s-wave wave velocities ($V_{p}$ and $V_{s}$), the dynamic elastic characteristics of specimens can be calculated as follows [35],
(2)$$\mu_{d}=\frac{{\left(\frac{V_{p}}{V_{s}}\right)}^{2}-2}{2\left[{\left(\frac{V_{p}}{V_{s}}\right)}^{2}-1\right]}$$
(3)$$E_{d}=\rho V_{p}{}^{2}\frac{\left(1+\mu_{d}\right)\left(1-2\mu_{d}\right)}{1-\mu_{d}}\times 10^{-3}$$
where $\mu_{d}$ is the dynamic Poisson’s ratio; ρ is the density of the specimen, g/cm^3^; $V_{p}$ is longitudinal wave velocity, m/s; $V_{p}$ is Shear wave velocity, m/s; $E_{d}$ is dynamic Young’s modulus/GPa. The testing procedures are shown in Figure 2.

#### 2.4.3. Compressive and Splitting Strength Tests

The compressive and splitting strength test was carried out using the WHY-2000 microcomputer-controlled pressure testing machine (Quanli, Jinan, China), according to GB/T 50081-2016 [36]. At the beginning, the cylindrical HPC specimens were used to measure the compressive strength. The loading mold of the uniaxial compression was stress controlled at a loading rate of 0.5 MPa/s until the specimen failed, and the failure stress was recorded as the peak compressive stress. Subsequently, the lower pressure plate of WHY-2000 was replaced by a curved steel pad clamp with a radius of 75 mm, and the cubic specimens were used for the splitting strength test. The loading mold of the uniaxial compression was stress control at a loading rate of 0.5 MPa/s until the specimen failed, and the failure stress was recorded as the peak splitting stress of specimens [37]. The compressive and splitting procedures are shown in Figure 3. In addition, these mechanical parameters were also applied to calculate the elastic modulus of specimens, and compared to the data obtained from ultrasonic velocity.

#### 2.4.4. Scanning Electron Microscope (SEM)

A JSM-7800F (JEOL, Tokyo, Japan) equipped with an energy-dispersive spectrometer (EDS) was applied to the microscopic morphology imaging and chemical composition observation. The specimens were processed into flat sheets with diameter of approximately 5 mm, and dried in an oven at 105 °C for 24 h. The dried sheets were attached to the carrier table in sequence and then placed in the equipment for vacuum processing and photo imaging. The accelerating voltage used for image acquisition is 10 kV, and dead time for chemical composition analysis is 1024 s.

#### 2.4.5. Nuclear Magnetic Resonance (NMR)

A MesoMR23-60 nuclear magnetic resonance (NMR) analyzer (Newmark Technology, Suzhou, China) was used to detect the hydrogen proton attenuation signal in the pore and analyze the Carr–Purcell–Meiboom-Gill (CPMG) pulse sequence (magnet strength 0.55 T, H-proton resonance frequency 23.320 MHz, magnet temperature 32 °C). Then, the pore structure characteristic parameters of specimens were collected by means of the CPMG. Before the test, cylinder specimens were saturated with deionized water for 1 h, with surfaces then wiped with solution, and subsequently wrapped with plastic film to avoid water evaporation.

The NMR transverse relaxation time ($T_{2}$) is divided into three parts, namely the bulk relaxation time ($T_{2bulk}$), the surface relaxation time ($T_{2s}$), and the diffusion relaxation time ($T_{2D}$, ms). The relationship is as follows:(4)$$\frac{1}{T_{2}}=\frac{1}{T_{2bulk}}+\frac{1}{T_{2s}}+\frac{1}{T_{2D}}$$

Bulk relaxation is an inherent characteristic of the fluid, which is determined by its chemical composition and viscosity. Surface relaxation occurs on the surface of rock particles and is related to surface wetness and pore size. Diffusion relaxation is determined by the fluid diffusion coefficient, the magnetic field gradient, and echo interval. Therefore, the formula can be further expressed as,
(5)$$\frac{1}{T_{2}}=\frac{1}{T_{2bulk}}+\rho \frac{C}{r}+D\frac{{\left(\gamma GT_{E}\right)}^{2}}{12}$$
where ρ is the surface relaxation coefficient, μm/s; r is the pore radius, μm; C is a constant, related to the pore shape; D is the diffusion coefficient, cm2/s; G is the magnetic field gradient, Gs/cm; γ is the gyromagnetic ratio, Hz/Gs; $T_{E}$ is the echo interval, s.

The diffusion relaxation term can be ignored for both saturated water and saturated oil conditions when the NMR response is observed at a small echo interval (e.g., 0.1 ms). Then, the equation transformed into,
(6)$$\frac{1}{T_{2}}=\frac{1}{T_{2bulk}}+\rho \frac{C}{r}$$

## 3. Results and Discussion

### 3.1. Mass Change

The mass change of HPC specimens at a predetermined time after sulfate corrosion was extracted to the mass loss rate curves, as exhibited in Figure 4. After soaking in sulfate solution for 100 days, the mass loss of unmodified HPC specimen is unobvious, but subsequently increased after 200 days of soaking, and accelerated at 200–300 days. The incorporation of nano-silica (NS) decreased the mass loss, especially at early ages. It can be seen that after soaking for 100 and 200 days, the NS additives even slightly increased the mass of HPC specimens, and this effect was augmented with higher NS addition. However, the long-term effect of NS seemed opposite to that at the early ages. After soaking for 300 days, the greatest mass loss was obtained in the 5% NS specimen, which is 26.5% larger than that of unmodified specimen (4.421% vs. 3.494%), whereas the 1% NS specimen decreased the mass loss by 13.5% in comparison to the unmodified specimen (3.023% vs. 3.494%).

In a comparison of surface views of specimens before and after soaking (Figure 5), it can be seen that the sulfate attack damaged the surface construction of HPC specimens, led to the porosity formation, and white crystalline precipitates filled in. The addition of NS alleviated these damages. For example, after soaking for 200 days, the pores on the surface of 1% NS HPC specimen are smaller than that on the modified HPC specimen. In addition, there are fewer precipitates in the porosity on the surface of 1% NS specimens than on the unmodified specimen after soaking for 300 days.

The mechanism of sulfate attack on concrete has been well established. When gypsum reacts with the aluminate-baring phases, the ettringite crystals form and these expansion products further damage concrete structures, due to their much greater molar volume than the aluminate-baring phases [1]. Therefore, to limit the content of C_3_A and C_3_S is one of the most effective methods for improving the durability of concrete when subjected to sulfate attack. The pozzolanic activity of NS can reduce the calcium hydroxide content during the C_3_S and C_2_S hydration, then limit the ettringite formation [23,34]. On the other hand, the filling effect of pozzolanic activity of NS can also decrease the permeability of concrete and mitigate the microstructure damage in front of sulfate attack [38,39,40]. Therefore, the mass of NS modified specimens even slightly increased after soaking for 200 days. However, these effects have a considerable decrease after long-term soaking, and the most effective dosage was found with 1% nano-NS replacement. The high dosage (mostly more 3 wt%) of nanoparticles replacement of cement was reported to induce an agglomerate tendency, which prevents the homogenous hydrate microstructure formation and leads to weak zones, contributing to the reduction of mass after long-term sulfate attack.

### 3.2. Elastic Modulus

The elastic modulus obtained from ultrasonic waves and strain–stress curves are exhibited in Figure 6. It can be seen that both overall trends of elasticity are consistent with each other, where the initial values are roughly the same among specimens. Then, the elastic modulus gradually decreases with increasing soaking time. The 1% NS and 2% NS modified specimens had greater elastic modulus at every time-point, whereas the 3–5% NS addition led to a reduction in elastic modulus in comparison to the unmodified specimens. Especially, after 300 days of soaking, the elastic modulus was elevated by 50.4% in 1% NS HPC specimens compared to the plain HPC specimen (34.9 vs. 23.2 GPa).

A proper amount of nano silica can make full use of the space resistance property of polycarboxylic acid water reducer and produce a synergistic effect, enhancing the elastic modulus of HPC specimens [22,41,42]. Hence, it can be seen that the elastic modulus is obviously elevated in NS modified specimens, especially with 1% NS additive. Too much nano silica can easily induce agglomeration and generate the hardening components in cement, which cannot be arranged in an orderly manner, leading to a gap between the hardening components in the concrete [41,43,44].

### 3.3. Compressive and Splitting Strength

Figure 7 illustrates the variations of peak compressive strength of HPC specimens after soaking for pre-determined periods. In unmodified HPC specimens, the compressive strength gradually reduced after soaking for 100 days and 200 days, and then rapidly decreased after 300 days. The NS addition alleviated the reduction in compressive strength after soaking, especially after 300 days, but was concentration and time dependent. The compressive strength was consistently greater in 1% NS HPC specimens in comparison to the unmodified HPC specimen, which increased by 8.8%, 17.6%, 21.5%, and 31.7% after soaking by 0 day, 100 days, 200 days, and 300 days, respectively. Although the compressive strengths were mostly higher in 2–5% NS HPC specimens compared with unmodified specimens at every time-point, they were all less than that of 1% NS specimens. After soaking for 200 days, the compressive strengths of 2–5% NS HPC specimens was even smaller than that of the unmodified HPC specimen.

The variations in splitting strength are similar to that of compressive strength, except for a slight increase after soaking for 100 days. As shown in Figure 8, the peak splitting strength of HPC specimens slightly increases after the first 100 days of soaking, and then decreases with increasing soaking time. After soaking for 300 days, it decreased by 37.2% in unmodified specimens (4.3 vs. 5.9 MPa). Consistent with compressive strength, the 1% NS additive is the most effective dosage in front of sulfate attack, which increased by 10.2%, 16.7%, 34.5%, and 69.8% after soaking for 0 day, 100 days, 200 days, and 300 days, respectively. The splitting strengths were mostly higher in 2%–5% NS HPC specimens compared with unmodified specimens, but they were all less than that of 1% NS specimens.

With respect to the compressive strength, the pozzolanic activity and filling effect of NS can promote the strength of cement-based materials [21,26,41,45]. Therefore, the quality of HPC specimens can be guaranteed under the conditions of sulfate attack. On the contrary, with the addition of too much powder, the nanoparticles in the cement matrix are difficult to evenly disperse, and the water consumption in the hydration reaction process weakens the working performance of concrete, resulting in an increase in the number of micro cracks and weak areas of concrete [7,46]. Therefore, it is difficult to guarantee the quality of concrete test block under sulfate attack.

Meanwhile, for the splitting strength, the dispersed NS in HPC can provide a large number of attachment points for the cement hydration crystallization in the concrete, thus improving the corrosion resistance, and further improving the splitting strength of concrete [20,27,47]. However, too much silica powder will easily induce the polymerization of HPC, making the nano molecules difficult to evenly disperse, and leading to the increase of micro cracks and weak areas of concrete, such that the splitting strength of concrete decreases.

### 3.4. T_2_ spectrum and Pore Size Distribution

The *T_2_* spectrum detected by NMR can transform and judge the cumulative evolution law of HPC in response to sulfate erosion. As shown in Figure 9, the overall trend of *T_2_* spectrums are bimodal curves with major and minor amplitude peaks. The major peaks are mainly distributed in the micropores range (<10 ms) and the minor peaks are usually in the mesopores range (10–100 ms), indicating that the porosity of HPC specimens is mainly micro- and meso-pores. Sulfate erosion obviously promotes the porosity evolution of the unmodified HPC specimen, which initially increases the number of micropores, and then affects the meso- and macro-pores with increasing soaking time.

The mixture of NS in HPC specimens changes the initial porosity distribution and the further effect of sulfate attack on porosity evolution. For the initial porosity, the addition of NS leads to the major and minor amplitude peaks of *T_2_* spectrums shifting to the left, and the major amplitude peaks are magnified while the minor peaks are reduced with increasing NS concentration. When the concentration of NS reaches 5%, the *T_2_* spectrum even transforms into unimodal. These results indicate that the addition of NS decreases the initial size of porosity, and this effect is magnified with increasing concentration.

Sulfate attack also leads to an increase of porosity in the NS modified HPC specimens, but the degree to which this occurs obviously varies with soaking time and NS concentration. When the soaking time is less than 200 days, the increases of amplitude peaks are similar in NS modified specimens (1–5%), which are smaller than the unmodified specimen. However, after 300 days of soaking, the *T*_2_ spectrum of NS modified specimens shift to the right and the minor amplitude peaks obviously increases, reflecting the increase of meso- and macro-pores in HPC specimens, and this effect is also magnified with increasing concentration. Notably, after 300 days of soaking, the increments of porosity in NS modified specimens are even greater than that of the unmodified specimen when the concentration is more than 2%.

Consistent with previous conclusions [3,20,21,48], the pozzolanic activity and filling effect of NS significantly change the porosity distribution of HPC specimens. The amount of micropores increased whereas the meso- and macro-pores decreased in the NS modified HPC specimens, and these changes are more obvious with higher NS concentration. This micro-filling effect may also block the sulfate ions penetration when subject to sulfate attack [7,49,50]. Therefore, the sulfate attack led to less porosity promotion in NS modified HPC specimens than in plain HPC specimens.

### 3.5. Micro-Surface Morphology

The micro-surface morphologies observed by SEM reflect the microstructure changes of the HPC specimen response to sulfate attack. As shown in Figure 10, there are a lot of micropores, mesopores, and macropores distributed on the micro-surface of the unmodified HPC specimen, due to the curing of HPC. Meanwhile, for the NS modified specimens, there are a large number of white crystalline filled in pores, in that the micropores are predominant on the micro-surface, and this phenomenon is more obvious with increasing of NS concentration.

After soaking in sulfate solution for 300 days, a large number of macropores and cracks are observed on the micro-surface of unmodified HPC specimen, whereas on the NS modified specimens, there are fewer macropores and cracks on the micro-surface, due to the increased white crystals in the pores. However, this inhibition effect also concentration dependent. As shown in Figure 10, when the concentration is 1%, there are mainly micropores on the micro-surface of specimen, accompanied by less crack formation than that of unmodified HPC. However, with the concentration of 2–5%, there are more macropores and obvious cracks formed after soaking, even more than that of unmodified HPC. EDS results are available in the Appendix A.

These results indicated that the filling effect of NS will significantly reduce the accumulated porosity, which optimized the concrete interface structure. However, the agglomerate at high NS dosage led to the coarser and more heterogeneous micro-surface of concrete after sulfate attack. These findings are also consistent with previous studies [21,45,47,51,52].

Considering that the negative effects of a high concentration of nano-particle additives were mostly attributed to the agglomerate, the ultrasonic mixing was introduced to prepare the nanoparticles, for better dispersion and to prohibit the further agglomeration. As Hamed et al. reported that at a relatively high percentage (5%, 7.5%, and 10%) replacement of cement by NC, the sonic mixing improved the mechanical properties by 1.4–3.7 times compared to that by as-received NC mixes [26] However, some researches did not support the effectiveness of “ultrasonic mixing”. As Faried et al. reported, the pre-mix of nano-particles in water demonstrated negligible differences in physical and mechanical properties in comparison to the as-received mixture, regardless of the mixture time [39]. Shaikh et al. concluded that the dry mixing of nano silica performed better than that with ultrasonic mixing [51]. Therefore, in the present study, we did not apply the “ultrasonic mixing” method, but it will be tested in future works.

## 4. Conclusions

It is of significance to improve the resistance ability of high-performance concrete (HPC) subject to sulfate attack. In this study, the performance of HPC mixed with nano-silica was investigated, and some useful conclusions were obtained as follows:(1)The sulfate attack promotes the porosity evolution of HPC, which leads to the decrease of mass, compressive, and splitting strength, resulting in the decline of strength and durability of HPC.(2)The mixture of nano-silica (NS) in HPC will change the porosity abundance and size, then inhibit the sulfate corrosion on HPC, but this effect is time and concentration dependent.(3)HPC specimens modified with the concentration of 1% NS exhibit the most effective inhibition of sulfate corrosion, where the mass loss, porosity increment, as well as compressive and splitting strength reduction are minimal after sulfate solution soaking.(4)The resistance effect response to sulfate attack can be attributed to the pozzolanic activity and filling effect of NS, which fill in the pores and further block sulfate ion infiltration. The agglomeration of nanoparticles at high concentration may be attributed to their decline of effectiveness.

## Figures and Tables

**Figure 1 materials-15-07614-f001:**
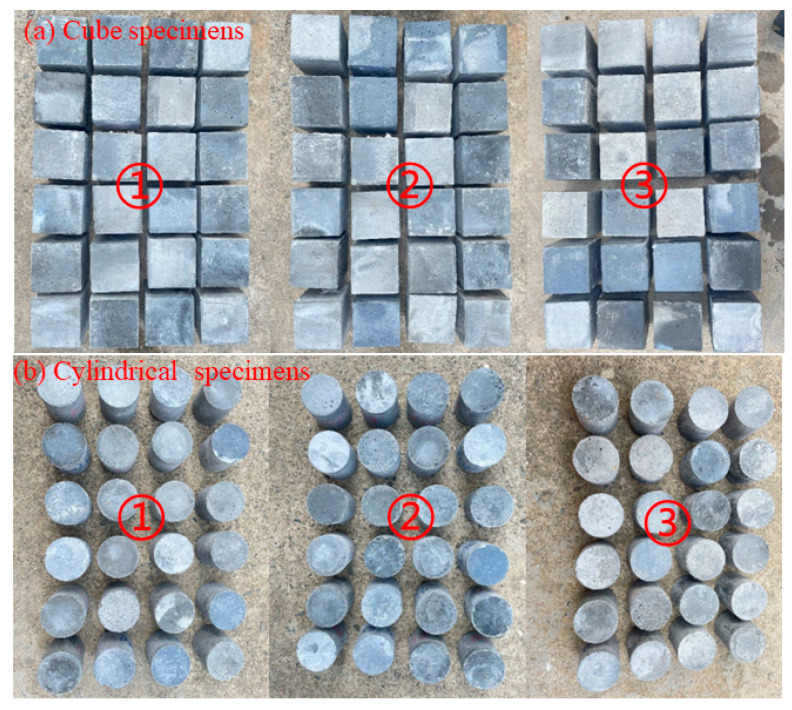
The cubic and cylindrical HPC specimens. (**a**) cube specimens, ① group 1, ② group 2, ③ group 3, (**b**) Cylindrical specimens, ① group 1, ② group 2, ③ group 3.

**Figure 2 materials-15-07614-f002:**
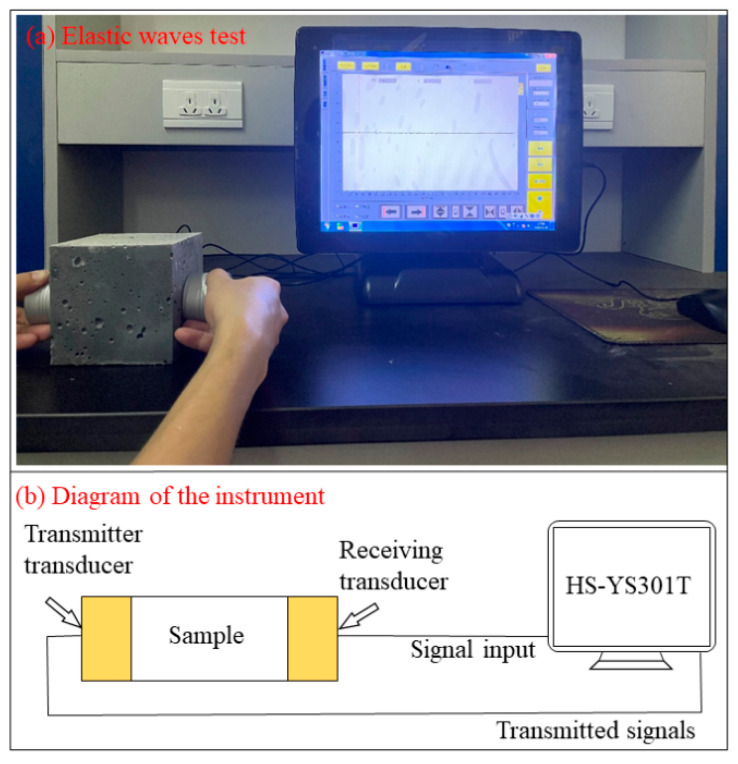
The photo (**a**) and diagram (**b**) of elastic wave test procedures.

**Figure 3 materials-15-07614-f003:**
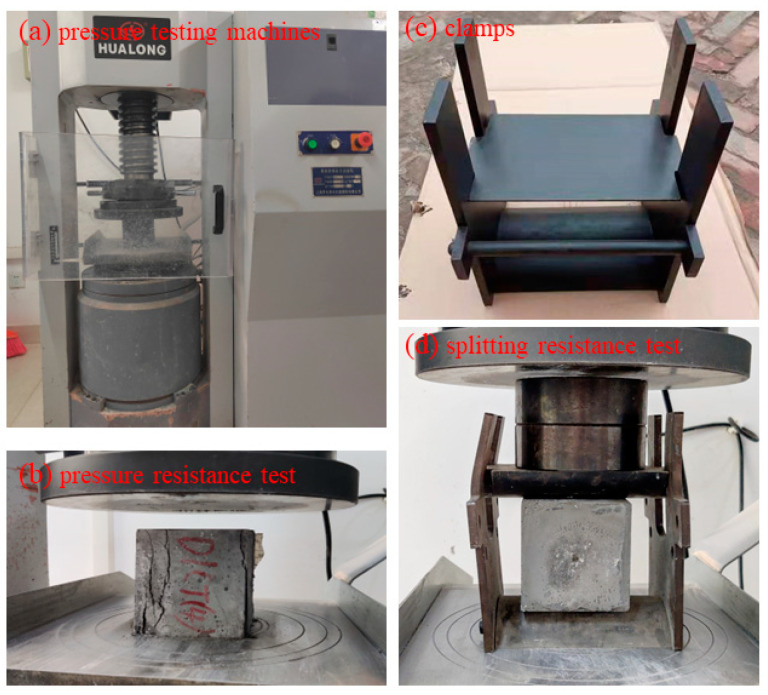
The compressive and splitting strength tests. (**a**) the loading mold of uniaxial compression; (**b**) the loading procedures of compressive strength test; (**c**) the pad clamp for splitting strength test; (**d**) the loading procedures of splitting strength test.

**Figure 4 materials-15-07614-f004:**
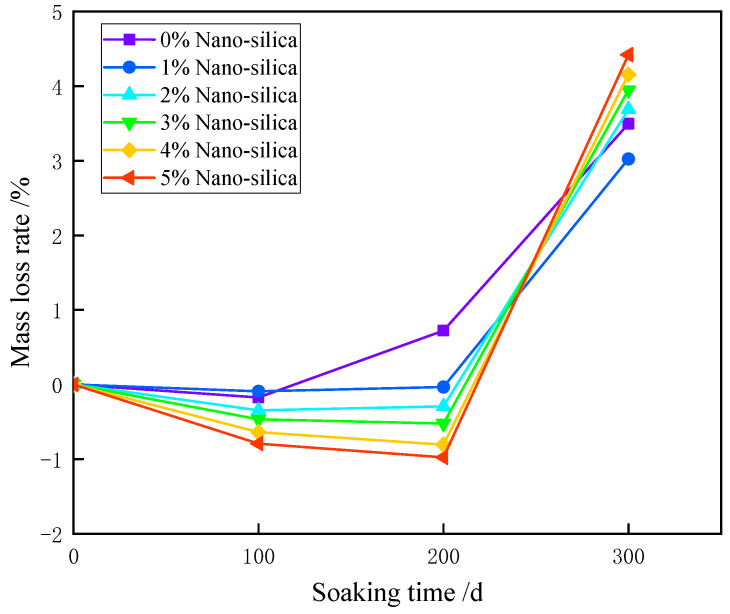
The time-dependent mass loss of HPC specimens subject to sulfate erosion.

**Figure 5 materials-15-07614-f005:**
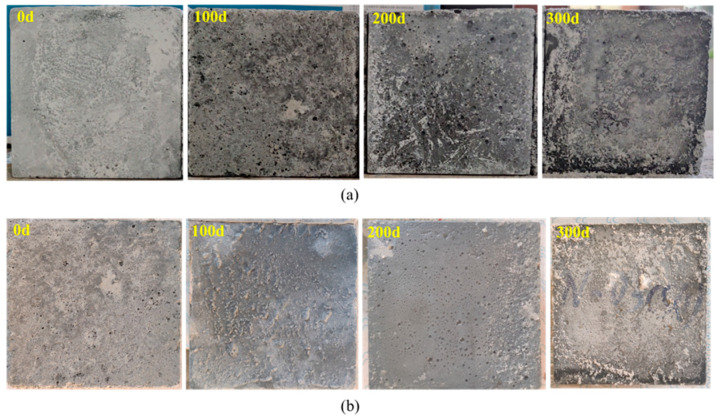
Concrete surface subject to sulfate attack: (**a**) 0% Nano-silica HPC; (**b**) 1% Nano-silica HPC.

**Figure 6 materials-15-07614-f006:**
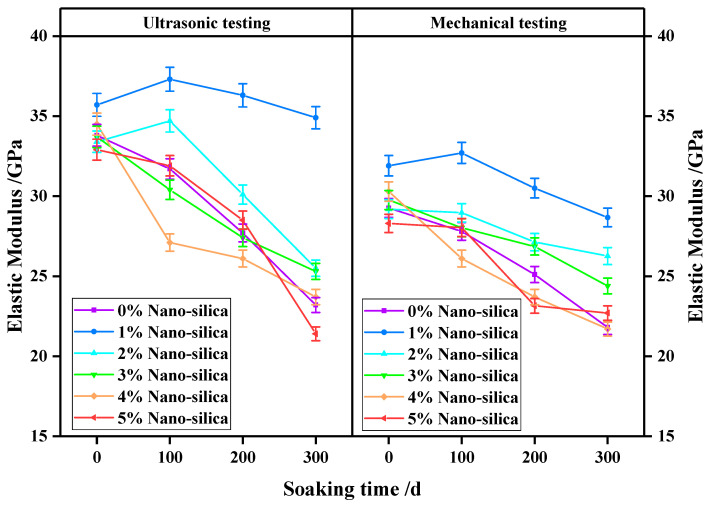
The time-dependent elastic modulus of HPC specimens obtained from ultrasonic and mechanical testing subject to sulfate attack.

**Figure 7 materials-15-07614-f007:**
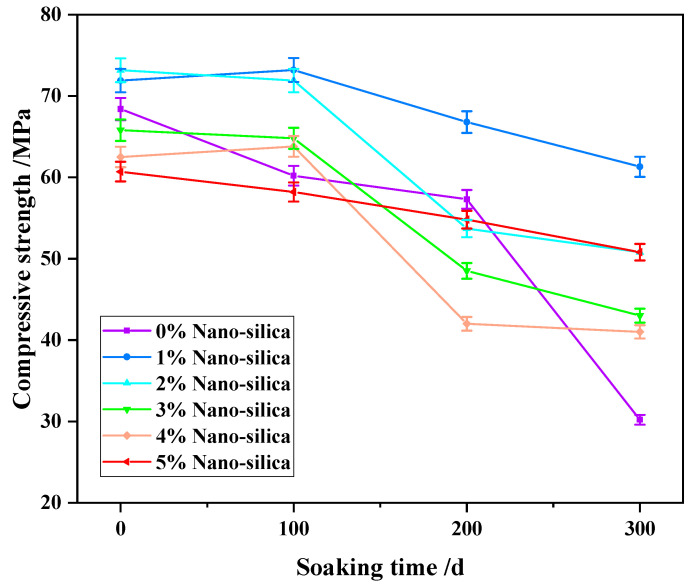
The time-dependent compressive strength of HPC specimens subject to sulfate attack.

**Figure 8 materials-15-07614-f008:**
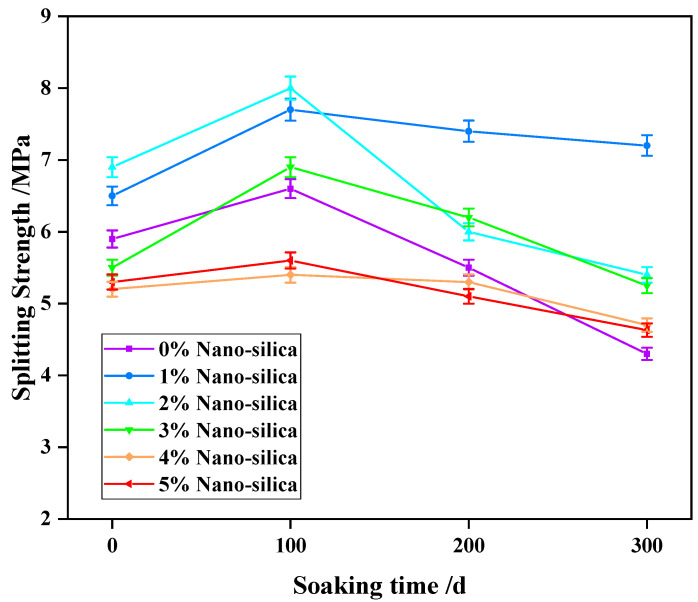
The time-dependent splitting strength of HPC specimens subject to sulfate attack.

**Figure 9 materials-15-07614-f009:**
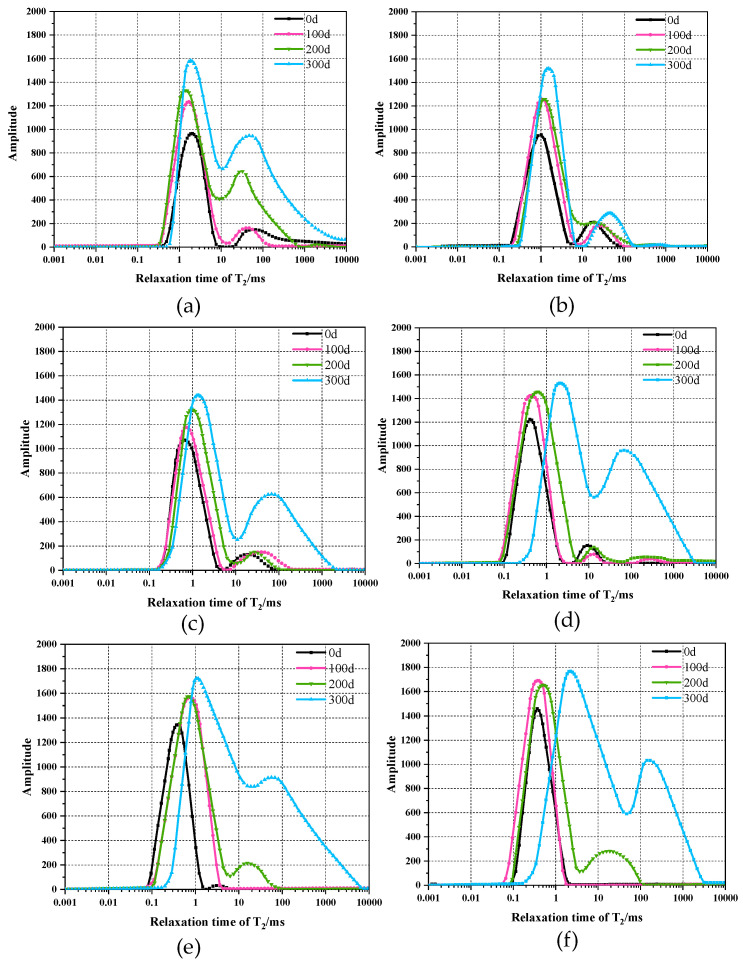
The T2 Spectrum and pore size distribution of HPC specimens subject to sulfate erosion: (**a**) 0% Nano-silica; (**b**) 1% Nano-silica; (**c**) 2% Nano-silica; (**d**) 3% Nano-silica; (**e**) 4% Nano-silica; (**f**) 5% Nano-silica.

**Figure 10 materials-15-07614-f010:**
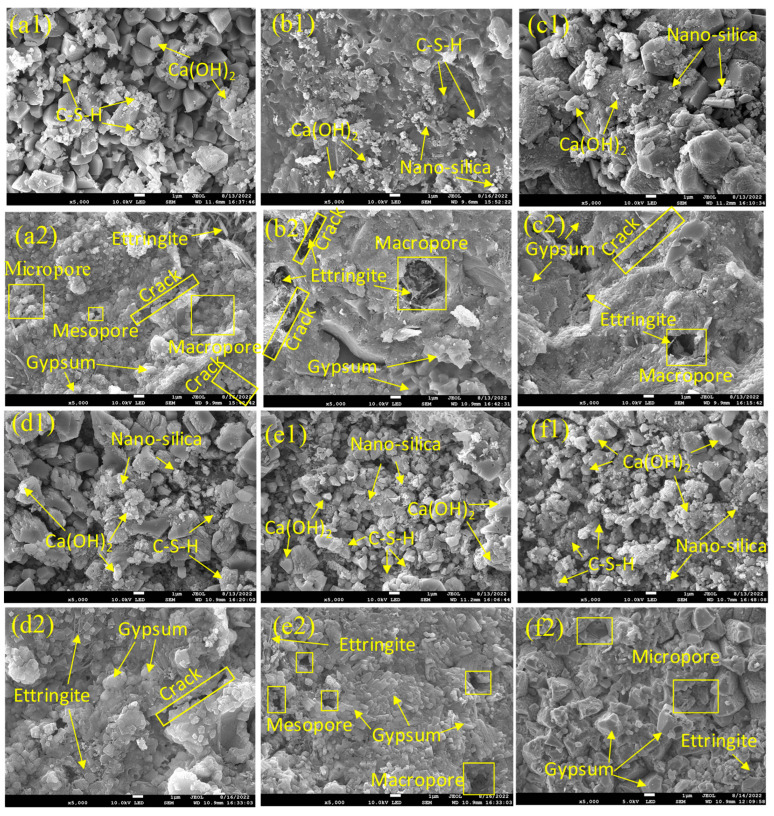
The SEM images of HPC specimens before (**a1**–**f1**) and after (**a2**–**f2**) sulfate attack. (**a1**,**a2**), 0% Nano-silica; (**b1**,**b2**), 1% Nano-silica; (**c1**,**c2**), 2% Nano-silica; (**d1**,**d2**), 3% Nano-silica; (**e1**,**e2**), 4% Nano-silica; (**f1**,**f2**), 5% Nano-silica. The minerals were determined based on the EDS chemical analysis.

**Table 1 materials-15-07614-t001:** The chemical compositions of cementing material (wt %).

Material	Chemical Compositions								Mineral Composition			
	SiO_2_	Al_2_O_3_	CaO	Fe_2_O_3_	MgO	SO_3_	Na_2_O	K_2_O	C_3_S	C_2_S	C_3_A	C_4_AF
Cement	21.34	4.84	63.61	3.43	1.26	2.45	0.16	1.12	48.19	22.64	7.03	10.43
Fly ash	69.17	6.94	10.52	7.47	1.16	0.11	2.13	1.26	-			
Silica fume	98.08	0.34	0.12	0.04	0.34	0.42	-	-				

**Table 2 materials-15-07614-t002:** The physical properties of Nano-Silica.

Size (nm)	Specific Surface Area (m^2^/g)	Volume Density (g/cm^3^)	Density (g/cm^3^)	Color
20	240	0.06	2.2–2.6	white

**Table 3 materials-15-07614-t003:** Mix proportion of high performance concrete (g).

Number	Cement	Fly Ash	Silica Fume	Quartz Sand	Water	Water-Reducer	Nano-Silica
NS-0-m-n	900	50	50	1100	250	8	0
NS-1-m-n	891	50	50	1100	250	8	9
NS-2-m-n	882	50	50	1100	250	8	18
NS-3-m-n	873	50	50	1100	250	8	27
NS-4-m-n	864	50	50	1100	250	8	36
NS-5-m-n	855	50	50	1100	250	8	45

Note: NS-0-, NS-1- etc., the content of nano-silica; m, the number of days of immersion (0, 100, 200, 300 days); n, the parallel test number (1, 2, 3). As the size of the test specimen for compressive and splitting strength is 100 mm × 100 mm × 100 mm, the data in Table 3 is the mass of each component per 1000 cm^3^ (one specimen), and the unit is g.

## Data Availability

The data presented in this study are available in the article or Appendix A.

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
