# Peer review of "Analysis of High Performance Concrete Mixed with Nano-Silica in Front of Sulfate Attack"

_materials, 2022, doi:10.3390/ma15217614_

Round 1

Reviewer 1 Report

The authors have presented an interesting study on the sulfate resistance of HPC concrete containing nano-silica. The study clearly addresses a limitation with existing literature, and presents a comprehensive investigation of mechanical, physical and durability properties of NS HPC concrete. The paper is recommend to be revised based on the following minor comments before publication.

Comments

·       For the compression and tensile tests (and all other tests), it looks like only one specimen was tested for each soaking period and mix design. Normally, 3 specimens are tested for each specific parameter set and the final results averaged. The authors should discuss the reasoning behind examining only one specimen and the associated limitations with data variability.

·       The dynamic elastic modulus was obtained using an acoustic tester. Is it possible to compare this with the stress-strain data obtained during the compressive test?   

Formatting and language

·       The word ‘by’ should be removed from the title. Title should read as follows; Experimental Investigation on the Sulfate Erosion Resistance 2 Behavior of High Performance Concrete Mixed with Nano- silica

·       Line 11- the word ‘known’ is missing

·       Line 16- the word ‘respectively’ is missing at end of sentence.

·       Lines 103 and 106 – use proper subscript notation and not m_0

·       Figure 11 caption is hard to read. Please introduce a form of notation to label the tests and make the caption more readable.

·       Overall, a thorough proof-reading is required to address typographical errors.

Reviewer 2 Report

The paper provides information about the influence of nano-silica on the resistance of the high performance concrete to sulfate erosion. The comprehensive laboratory investigation were determined covering durability and mechanical properties. There are only some issues that should be elaborated. The comments to the paper are presented above:

- The authors haven’t presented the views of the specimens after being subjected to sulfate erosion. It would enrich the paper;

- The results from mechanical properties tests shown in Figures 7-9 should be presented in numbers in table. The CV (coefficient of variation) should be determined.

- Point 2.3.4 – The split tensile strength was determined based on the Chinese standards. Could you please provide more information about it? According to the European standards it is performed on cylinders;

- Table 3 – Please revise units;

- Figure 11 – Please revise the title of the figure to avoid repetition of words:” Nano-silica mixed concrete of”.

Reviewer 3 Report

REVIEWER´s Indications, Clarifications, Suggestions, Proposals and Comments 

1. The Title is not appropriate because commonly, the sulphate-resistance of Portland cements with or without pozzolan-type active mineral additions (NS in this case), is not determined by the erosion (or loss of mass) of their mortar or concrete specimens. conventional or high mechanical resistance, but by:

1.1. The loss of its volume stability determined by the greater or lesser expansion suffered by its referred specimens with the course of the sulphate attack, which is in accordance with the lower and higher, respectively, sulphate resistance of the OPC with or without each of the pozzolans natural or artificial (NS in this case) that constitutes them (ASTM C 452-02 [1] and ASTM C 1912-95 [2], respectively, are appropriate for this purpose, although ASTM C 1912-95 expressly states in point 3.1. of its Section 3. Significance and use, that ASTM C 452-02 is not, but other study has shown that it is valid for differentiating cements resistant to sulphates from those that are not, only using another physical specification different from maximum allowable expansion at 28 days-age of its mortar specimens (ΔL28days ≤ 0.054%) [3], than that used for Portland sulphate-resistant cements (SRPC) at 14 days-age (ΔL ≤ 0.040%) [4], and

1.2. The consequent decrease or no greater or lesser of its compressive strength that occurs along with said sulphate attack, to its corresponding mortar or concrete specimens (those manufactured and tested by the authors are appropriate) but only when the Pozzolanic addition hinders and even prevents the attack of sulphates in an amount appropriate to the C3A (%) content of the OPC with which it has been mixed, in the case of NS (artificial silicic pozzolan in chemical character such as microsilica and diatomites) used by the authors because if the chemical character is totally opposite, aluminic, like that of metakaolin, compressive strength is not needed logically but only ΔL(%) vs. Time [5] or ΔØ vs. Time. 

Because the cement paste of the specimens and, more specifically, the C3A content (%) of the OPC with which it has been mixed and which the NS will protect from said sulphate attack in an amount appropriate to said C3A content, is the one that really suffers from sulphate attack. Which, on the other hand, forces us to determine the potential mineralogical composition of the Portland cement used by the authors, by means of the Bogue formulas [11], or real, by means of its XRD analysis, Rietveld method. And for its potential mineralogical composition, the OPC needs to determine the content of many more chemical parameters than the 5 that they have determined alone. In addition to its total content of SiO2 [12-13], and CaO free [15]. While for NS it is necessary to determine its total SiO2 (%) [16] and reactive SiO2 (%) content [13 or 17], its pozzolanicity by means of the Frattini test [18] (at 1, 2, 7 and 28 days-age) and its resistant activity index [16].

2. On page 1, Abstract, the authors have written the following: “Nano-silica (NS) is an effective material to improve the strength and durability of high 10 performance concrete (HPC), but little information is about its resistance to sulfate erosion"

Reviewer´s punctuation: It is not sulfate erosion but sulfate expansion vs. time that is necessary to determine.

Reviewer's suggestion: using gypsum instead of the 5% sodium sulphate solution used by the authors makes the results much easier and faster to obtain and, furthermore, makes them much more transcendent and meaningful for the user finish of the new Portland cement with NS resistant to attack by sulphates: civil engineers and architects.

Therefore, and in accordance with everything said by this Reviewer, the title of this article should be as follows: Performance of Nanosilica pozzolan and Portland cement in front of gypsum attack. (or in front of sulphates attack.)

Reviewer’s final comment: If the actual or potential C3A (%) content of the OPC used by the authors is high, replacement amount greater than 5% will surely be needed to prevent it sulfates attack and destroying it, justifying, incidentally, that lower amounts of replacement have not prevented it.

Additional new References

1.   ASTM C 452-02 Standard: Standard Test Method for Potential Expansion of Portland-Cement Mortars Exposed to Sulphate. ASTM International, 100 Barr Harbour Drive, PO Box C700, West Conshohocken, PA 19428-2959, EEUU.

2.   ASTM C 1202-95 Standard: Standard Test Method for Length Change of Hydraulic-Cement Mortars Exposed to a Sulphate Solution.

3.   Talero R. Performance of metakaolin and Portland cements in ettringite formation as determined by ASTM C 452-68: kinetic and morphological differences. Cement and Concrete Research, 35 (7); p. 1269-1284 (2005).

4.   ASTM C150/C150M-12 Standard. Standard Specification for Portland Cement.

5.   Talero R. Gypsum attack: performance of silicic pozzolans and Portland cements as determined by ASTM C452-68”. Adv Cem Res, 24 (1). p. 1-15 (2012).

11.    ASTM C150/C150M-12 Standard. Standard Specification for Portland Cement.

12.    ASTM C114-04 Standard. Standard Test Methods for Chemical Analysis of Hydraulic Cement.

13.    EN 196-2:2014 Standard. Method of testing cement. Part 2: Chemical analysis of cement. AENOR. Calle Génova, 6, 28004-Madrid, Spain.

14.    UNE 80210:1994 Standard. Methods of testing cements. Chemical composition determination of Portland clinker and cements by XRF. AENOR.

15.    UNE 80243:2002 Standard. Methods of testing cement. Chemical analysis. CaO free determination. Etilenglicol method. AENOR.

16.    EN 13263-1:2006 Standard. Silica fume for concrete. Part 1: Definitions, requirements, and conformity criteria. AENOR.

17.    UNE 80225:2012 Standard. Methods of testing cement. Chemical analysis. Reactive silicon dioxide determination in cements, pozzolans and fly ashes. AENOR.

18.    EN 196-5:2011 Standard. Methods of testing cements. Part 5: Pozzolanicity test for pozzolanic cements. AENOR.

Reviewer 4 Report

The influence of nano SiO(NS) content (1-5%) on the sulfate corrosion of high performance concrete (HPC) is analysing in the paper. In the paper is presented interesting research results, that 1% of NS is the best for HPC, but it could be only because of bad NS particles separation and agglomeration. Authors could use other methods for preparation NS particles, e.g. ultrasound.

Remarks: In the Introduction must be added information about results of other research works. What amount of nano-materials was used and how significantly they change physical and mechanical characteristics of concrete (e.g. in %). It is written , that they are methods, how to decrease sulfate corrosion of concrete, but what properties of concrete were analysed, what results were presented in other works? 

Table 1. The content of numbers after come should be the same. How was established chemical composition?

Table 3. It is written, that composition is presented in kg/m3. Really? The composition must be corrected. How was calculated amount of water reducer? The main properties of water reducer should be presented.

2.4.2. Elastic modulus test. According to Figure 2 it was calculated ultrasound pulse velocity. The calculation formula should be presented. Why elastic modulus?

Figures 4-5 should be deleted, because they do not provide any relevant information.

How many samples was used for each test. The standard deviation should be presented. Why e.g. elastic modulus after 100 and 200 days is the lowest, when 2% of NS was used, but the best results are with 1% and 4% NS? Why such unevenness of results are established in all properties? The reason is only agglomeration of NS?

Figure 11. It is seen, that different crystal hydrates are shown in different batches. It should be presented EDS results.

357. "HMC"?

The paper could be published after major corrections.

Round 2

Reviewer 3 Report

ASTM standards are not really recommendations but test methods that must be put into practice as specified by each standard.

The potential mineralogical composition of the OPC (ASTM C150/C150M-12 Standard) used by the authors is needed to be able to know its potential C3A content (%) because the attack of sulfates is focused on it, which is difficult and even prevents if the amounts of SF and NS (mostly in this investigation), which are incorporated into the concrete are consistent with said content of C3A. And this last good behavior is the one that needs to be revealed so that the physical parameter ΔL (%) vs. Time is much more important and significant than the erosion used by the authors.

The chemical parameters in Table 1 are not sufficient to determine the potential mineralogical composition of the OPC used by the authors because said ASTM C150/C150M-12 Standard indicates many more.

Table 3: In the opinion of this Reviewer, Table 3 is paradigmatic because:
1. First and foremost, all SF, NS and FA are the same for the authors. And although this premise can be admitted/accepted to a certain extent for SF and NS, for FA it cannot be admitted/accepted in any way or under any circumstances. Because its variability in chemical, mineralogical, morphological composition and average size of spherical, hollow and more or less vitreous particles is enormous, for which reason, first of all, it is necessary to determine its chemical character as a result of its corresponding contents of reactive silica and reactive alumina. especially and above all. And depending on these contents, their chemical character will be more aluminum than silicic or contrary to what is usually the most common, which makes them somewhat closer to SF and NS, whose chemical character is totally silicic, than to MK whose chemical character is totally contrary or opposite: aluminic, and
2. In 2nd place, the dosages NS-4-m-n and NS-5-m-n contain the percentage contents of SF+NS closest to the maximum allowed replacement amount of SF by OPC: 10%, which is why the dosage maximum of ≈ 5% of NS has been the one that has shown the worst behavior of all in each and every one of the tests to which the authors have subjected their concrete for having generated and developed their dosages of FA + SF + NS the highest pozzolanic activities of the 6 dosages analyzed and studied by the authors. Said pozzolanic activity had to have been determined, first of all, by means of standard EN 196-5 or Frattini test (which is easy and simple to perform and inexpensive to implement) and since the [CaO] and [OH–] of the liquid phase of their pastes would have been the lowest of all the dosages tested, the last two could not have been considered or tested without further ado.
3. In addition, having replaced the NS with OPC contributes to the previous behavior seen in point 2 and verifies, on the other hand, that said replacement has not been appropriate or logical because the most logical thing would have been to replace it with FA, which, with total certainty , is the least pozzolanic of all.

All of which justifies, ultimately, that the dosages in Table 3 are not the most appropriate for the objective intended by the authors due to lack of the necessary rationality because they are not specialists in pozzolanic materials.

Results and Discussion
3.1. Mass change

Reviewer´s clarification: The last 3 paragraphs of this Section are much more about supposition and speculation than verification of behavioral hypotheses. In research, behaviors should not be assumed but rather verified, which is the best way to justify them.

And the height of that assumption is precisely the last sentence of the last paragraph. And it is that the justification that the authors give to the loss of mass with the increase of the NS content is wrong because this final bad behavior of ≈ 5% NS in the test tube is due to the very high pozzolanic activity that it has generated and developed. at such high ages of sulfate attack (which could have been determined very easily using the EN 196-5:2011 Standard, or even more easily: determining the pH of the liquid phase of the OPC cement paste with ≈ 5% NS ) which makes cement-based materials unstable without further ado. On the other hand, and at such advanced ages of sulfate attack, C3A is not generated, but is transformed into slowly forming ettringite [17,18], but with an adequate amount of SF+NS it is not transformed because it prevents it. .

All of which denotes the notable lack of experience and knowledge of the authors in OPC-based construction materials, in general, and in their attack by sulfates, very specifically. To the point of not knowing how to justify their behavior as it should.

1.1.  Mass change and 3.2. Elastic modulus

Reviewer´s clarification: Section 3.1. The justification for the behavior is wrong and the last paragraph of Section 3.2. is, once again, an assumption or speculation of the authors and, furthermore, also erroneous because it is not due to the slow forming ettringite from the C3A (%) content origin of the OPC used but, once again, to the very high pozzolanic activity that SF, FA and NS (5%) have generated and developed at such high ages of sulfate attack.

Reviewer´s clarification: Section 3.1. The justification for the behaviour is wrong and the last paragraph of Section 3.2. and that of Section 3.3. Compression and fracture strengths is, once again, an assumption or speculation of the authors and, furthermore, also erroneous because it is not due to slow forming ettringite or from C3A content (%) origin of the OPC used, but rather, once again, to the very high pozzolanic activity that SF, FA and NS (5%) have generated and developed at such high ages of sulfate attack.

Reviewer 4 Report

The quality of article was improved, but I have some remarks.

First word of title could be corrected to "Analysis" or "Research".

The marking in Fig. 10 is questionable. Please present EDS results in the Table. In some places, where is marked ettringite, it is looked as calcite, etc.
